# MODEL ENTANGLEMENT FOR SOLVING PRIVACY PRE-SERVING IN FEDERATED LEARNING

## ABSTRACT

Federated learning (FL) is widely adopted as a secure and reliable distributed machine learning system for it allows participants to retain their training data locally, transmitting only model updates, such as gradients or parameters. However, the transmission process to the server can still lead to privacy leakage, as the updated information may be exploited to launch various privacy attacks. In this work, we present a key observation that the middle layer outputs, referred to as data representations, can exhibit independence in value distribution across different types of data. This enables us to capture the intrinsic relationship between data representations and private data, and inspires us to propose a Model Entanglement(ME) strategy aimed at enhancing privacy preserving by obfuscating the data representations of private models in a fine-grained manner, while improving the balance between privacy preservation and model accuracy. We compare our approach to the baseline FedAvg and two state-of-the-art defense methods. Our method demonstrates strong defense capabilities against mainstream privacy attacks, only reducing the global model accuracy by less than 0.7% and training efficiency of 6.8% respectively on the widely used dataset, excelling in both accuracy and privacy preserving.

## 1 INTRODUCTION

Deep learning, particularly through the use of deep neural networks, has seen widespread adoption due to its exceptional performance, which is also heavily dependent on large volumes of high-quality training data. Currently, the widely adopted distributed learning algorithm known as Federated Learning (FL)McMahan et al. (2017) allows a central server to handle broadcasting and computation for the participating client nodes. This iterative process facilitates collaborative model refinement while preserving individual data privacy.

Although Federated Learning effectively mitigates the direct transmission of training data to enhance privacy protection, it does not provide security guarantees for clients' private local contributions. In related work, Zhu *et al.*Zhu & Han (2020); Geiping et al. (2020) demonstrated that even in scenarios where gradients are shared, adversaries could reconstruct training data. The scenario of sharing model parameters introduces additional privacy risks. Carlini et al. (2022); Li & Zhang (2021) utilized model outputs to conduct membership inference attacks, focusing solely on the structure of black-box models. These attacks aim to determine whether private data corresponds to the model's training data, effectively identifying membership.

Essentially, whether through reconstruction attacks or inference attacks, the core privacy threat stems from extracting the intrinsic relationship between the model and the underlying data. The evolving landscape of privacy attacks highlights the crucial need for developing robust strategies to protect sensitive information when sharing model parameters in distributed learning scenarios.

Currently, mainstream defense approaches against these attacks include Differential Privacy (DP), Secure Multi-party Computation (SMC), and Data Compression (DC). DP enhances privacy by introducing perturbations to shared data, though it often compromises model accuracy. SMC encrypts user data, allowing servers to aggregate encrypted information while preventing malicious eavesdropping, but it faces challenges related to key distribution and high computational demands. Encryption-based methods also require extensive matrix operations, leading to increased computational load and potential communication delays. While less common, DC similarly struggles with

balancing privacy defense and accuracy. Despite their strengths and limitations, ongoing research is essential to further optimize these methods for privacy in distributed learning. Therefore, it is imperative to propose a more comprehensive solution that fully leverages the collaborative nature of client training to achieve robust client-level privacy preservation.

Our preliminary experiments yielded a key observation: **when training data distributions differ, the model parameters and updates reflect these distributional variations, particularly in non-independent and identically distributed (Non-IID) datasets.** This phenomenon may be linked to the model's capacity to memorize data, which is a key factor in privacy attacks. Based on the above observations, this paper first investigates the intrinsic relationship between intermediate outputs of the model and the model updates(i.e, gradient information) with the privacy. Accordingly, a privacy measurement mechanism based on node importance is proposed. The results of privacy risk assessment are used as guidance to design a parameter replacement algorithm, which is then applied within a federated learning framework. This leads to the development of a federated multi-client collaborative privacy preserving framework with a safeguard client.

## 2 RELATED WORK

### 2.1 PRIVACY ATTACK

**Membership Inference Attack(MIA):** MIA enables an attacker to determine whether a sample $(x, y)$ belongs to the training dataset of a target machine learning model by constructing a binary classifier which output 1 or 0. Ye et al. (2022) analyzed various attacks and attributed the vulnerability of data points to different levels of memorization, or overfitting to conditional memorization. Recent research has explored MIA's significance in real-world scenarios. For instance, Chen et al. (2022) proposed a practical MIA against the industrial Internet of Things, relaxing key assumptions made by prior MIAs that were impractical in industrial settings. Zarifzadeh et al. (2024) introduced RMIA which is a refined MIA with lower overhead and uses fine-grained modeling of null hypotheses in likelihood ratio tests and achieves greater robustness and accuracy than existing methods.

**Model Inversion Attack:** Under the model inversion attack, the attacker attempts to restore the training data of the model with limited knowledge. Recently works, such as Zhu & Han (2020); Geiping et al. (2020); Zhao et al. (2020) had continuously improved this attack, making it more efficient. The research in Wang et al. (2022) analyzed how weight distribution affects the training data recovery from gradient and proposed the algorithm exploited the variance of gradients. Additionally, Nguyen et al. (2023) addressed the suboptimal loss functions and poor quality of reconstructed samples by introducing regularization terms to improve the loss function's convexity, thereby enhancing the accuracy of recovered samples.

### 2.2 PRIVACY PRESERVING TECHNOLOGY

**Secure Multi-party Computation:** MPC aims to compute private inputs from all parties through a secure function and return the result. Bonawitz et al. (2017) proposed HybridAlpha, a multi-party training method based on functional encryption, which introduced a trusted third party for verification. Zhang et al. (2020) developed Batchcrypt, a homomorphic encryption-based secure aggregation scheme for cross-organizational settings, which reduces communication overhead, though its performance significantly degrades with larger models compared to plaintext schemes. However, MPC methods incur substantial computational overhead and conceal model updates from the server, making them vulnerable to Byzantine and Poisoning attacks Wu et al. (2020); Chang et al. (2020). Chen et al. (2023) addressed issues related to detecting malicious parameters and the strong reliance on IID distributions in federated learning and MPC.

**Differential Privacy:** DP algorithms can resist corresponding privacy attacks by adding noise to the federated learning framework. The work Wei et al. (2020) introduced noise to local updates before the aggregation to defend against privacy attacks, optimizing the selection of the best K clients to balance privacy and efficiency.Zhu et al. (2022) proposed a fine-grained method to allocate noise according to the importance value of layers in order to remain high model performance. In address precision degradation, Wang et al. (2023) introduced the idea of dynamic defense in DP federated learning. However, existing DP strategies require clients to conform to a specific statistical distri-

bution collectively, so that their mutual noisy effects are neutralized after aggregation. In scenarios with fewer clients, the added noise significantly impacts individual client accuracy.

**Data Compression:** DC or pruning model updates (Tsuzuku et al. (2018)) is a practical approach to alleviate the connection between updates and private data. Zhu & Han (2020) achieved the effect of resisting DLG attacks by gradient information compresion and sparsification. However, this method requires manual setting of the compression ratio, and requires a higher compression rate for a better mitigation effect. To better address system heterogeneity and adapt to dynamic edge-client changes within the federated learning framework, some methods propose adaptive control of local updates for model compression. Miao et al. (2022) used compressive sensing and noise processing with a privacy budget, reducing the computational cost of differential privacy for large models. Similarly, Xu et al. (2023) optimized local updates based on client bandwidth and computational resources. However, these methods focus more on system convergence rather than demonstrating data compression's effectiveness in privacy preservation.

## 3 PRELIMINARIES AND PROBLEM STATEMENT

### 3.1 MODEL UPDATE

In this section, we consider a distributed joint training framework based on gradient aggregation. To better align with real-world scenarios, such as production federated learning, we opt to use model updates($\Delta g_n$) instead of individual gradients($\nabla g_n$) as the information uploaded by clients, as defined in Wang et al. (2023).

For a given participating client $C_n, n \in N$ with its local dataset as $D_n$, multiple rounds of local training are conducted on several mini-batches, with each epoch calculating an intermediate gradient. However, in practical scenarios, the results after local computations differ from the average gradient obtained across multiple epochs and mini-batches. In addition to fixed learning rates, hyperparameters such as momentum, weight decay, and learning rate schedules also need to be considered. Since the intermediate results of each epoch and mini-batch remain inaccessible to other clients and the server, we focus on the model's states at the beginning $w_n^t$ and the end of local training $w_n^{t+1}$. The difference between these states is uploaded as **model update** information for server aggregation $\Delta g_n^{t+1} = w_n^{t+1} - w_n^t$. The global model is then updated according to the aggregation result $W^{t+1} = W^t + \sum_{n=1}^{N} \frac{|D_n|}{|D|} \Delta g_n^{t+1}$.

### 3.2 THREAT MODEL

Since we assume the server is curious-but-honest, when clients transmit information to the server, it can construct a threat model through privacy attacks.This could also result in local privacy vulnerabilities. Specifically, the following privacy threats may arise.

**Membership Inference Attack:** The effectiveness of this attack hinges on the attacker's ability to access the target model, leverage existing information to obtain intermediate computation results, and use these to construct a binary classifier to determine whether a given sample belongs to the training dataset. More broadly, attackers may estimate the parameters of the target model through model updates. In the context of federated learning with gradient aggregation, after the client $C_n$ uploads the locally updated gradient $\Delta g_n^{t+1}$, the server directly uses it to update the global model $W^t$ from the previous round. This allows the reconstruction of the user's local model $w_n^{t+1}$ for the current round, enabling an curious-but-honest server to extract privacy information from the white-box model.

Formally, when the adversary is given a data point $z = (x, y)$, aiming to determine whether $z \in S$, where $S$ is the training dataset for the model $A^S$. The result $b = 0$ if $z$ belongs to $S$, and $b = 1$ otherwise. The adversary's output on the data point is denoted as $M$. The adversary's advantage $Adv^M$ is expressed as the difference between $M's$ true and false positive rates:

$$Adv^M = Pr[M = 0|b = 0] - Pr[M = 0|b = 1] \quad (1)$$

**Model Inversion Attack:** Privacy adversary can further utilize model update information such as gradients to reconstruct training data. For instance, in image classification tasks, attackers can reconstruct images pixel by pixel from gradients through optimization techniques(Zhu & Han (2020)). In

recent research, Geiping et al. (2020) proposed a model inversion attack method that approximates gradients using model updates. This approach employs a matching mechanism similar to traditional gradient matching, utilizing model updates and virtual increments to optimize target samples.

For the client $C_k$ with the training data $(x_k, y_k)$, the model update increment obtained after $t$ global rounds is denoted as $\Delta g_k^t$. The attacker attempts to reconstruct the training data by exploiting the increment. Specifically, the process begins by initializing a dummy data $(x', y')$, and then computing the virtual gradient as model update $\Delta g^*$ accordingly. The difference between the virtual gradient and the real gradient is then optimized, updating the dummy data $(x', y')$ to approximate the actual values based on the following objective:

$$x^*, y^* = \underset{x', y'}{\arg\min} \, Dist\left(\Delta g_k^t, \Delta g^*\right) \tag{2}$$

where $x^*, y^*$ represent the attacker's reconstruction results, and $Dist(\cdot)$ represents the distance function of the vector, such as L2 distance.

## 4 PROPOSED SCHEME

### 4.1 BASIC IDEA

We begin by introducing the model structure which can be roughly divided into convolutional layers for feature extraction and fully connected layers for classification. For simplicity, we consider an input image and a deep neural network (DNN) structure with one convolutional layer, one filter, and one fully connected layer as an example. We define the convolution layer and the fully connected layer as follows:

$$X = T(cir(W_c)R) \tag{3a}$$
$$r = W_f \cdot X \tag{3b}$$

where $R$ represents the raw input data to the convolution layer and $W_c$ denotes the convolution kernels, $X$ is the output of the convolution layer, which also serves as the input of the next fully connected layer. $cir$ refers to the circulant matrix and $W_f$ is the linear weight matrix. And we denote the intermediate output of layers as data representations $r$.

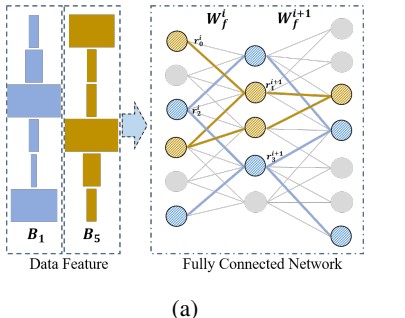
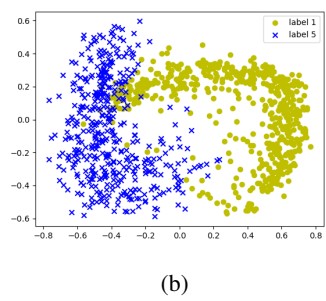

(a)                                                   (b)

Figure 1: (a)Data representation independence for samples with different labels. (b)Data representations distribution with labeled 1 and 5 after dimensionality reduction.

We observe that in fully connected layers, variations in the distribution of input features $X$ lead to corresponding changes in the distribution of intermediate output parameters within the layer after forward computation. We name it as the phenomenon of data representation independence for training data. As illustrated in Figure 1a, taking the samples with labels 5 and 1 as an example, the data representations $r$ of different classes are differentiated after the forward computation due to variations in the input features. Consequently, the computed data representations in the intermediate layer will also be relatively independent. This independence becomes even more pronounced when considering individual samples or complex model structure, where the data representation independence between samples is more apparent.

To verify that data from different classes correspond to distinct distributions of data representations, we selected 500 samples each with labels 1 and 5 from the

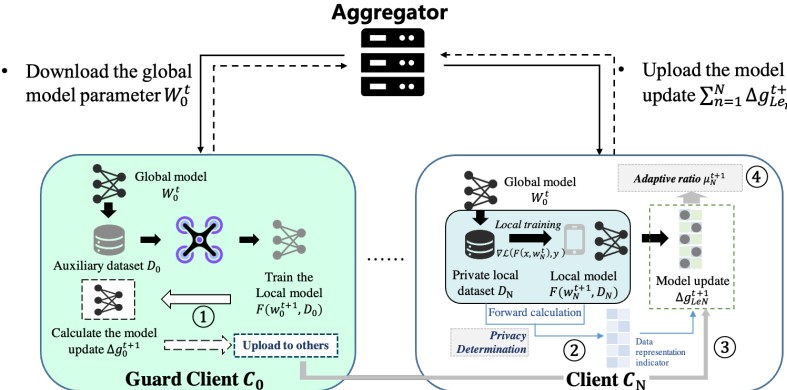

Figure 2: Work-flow diagram of the FL framework applying the ME strategy: (1) Local training for guard client; (2) Identify the location of parameters for private clients; (3) Set the replacement rules in descending order; (4) Calculate the update ratio and perform the aggregation.

MNIST dataset and input them into the LeNet5 model to obtain the data representations before the output layer. These representations were then visualized after dimensionality reduction, as shown in Figure 1b. The results clearly differentiate the data representations of the two classes, supporting our hypothesis regarding the independence of data representations. For a given model, features of different classes with distinct distribution cause the output data representations to occupy different corresponding positions in the output neurons, which introduces certain privacy risks.

This observation inspires us to develop a strategy that utilizes obfuscated data representations to reduce their independent performance and resist attackers, as illustrated in Figure 2. Specifically, to protect the private client $C_N$, our method first assesses the portion of the transmitted model update containing sensitive information by analyzing the process from data representation to gradient leakage and identifying the location of these parameters. These sensitive parameters of gradient are then replaced with an unrelated model from client $C_0$. To facilitate this, we introduce a guard model $F(W_0, D_{aux})$ which also acts as a participant in the FL scenario. The model is trained on the publicly available auxiliary dataset $D_{aux}$ as $D_0$ with minimal privacy concerns, such as data sourced from the Internet. The next private client receiving the model update will fine-tune it with local data and calculate the proportion of model update relative to historical information to account for the heterogeneity across different clients. The newly computed model update is then merged with the guard model update, reducing the privacy risk associated with the original model update. We will introduce the specific algorithm in detail in the following sections.

Our approach leverages the independence observation of data representation, introducing a interference model between clients with different training data distributions, and those parameters can also be regarded as a kind of noise information. This strategy aims to confuse the attacker's inference to protect privacy while minimizing the impact on model accuracy caused by replacement through FL aggregation.

## 4.2 DATA REPRESENTATION ENTANGLEMENT STRATEGY

### 4.2.1 DEFENSE STRATEGY

According to a previous research Sun et al. (2021), effective gradient optimization attacks can be launched using only the parameters of the fully connected layer. Consequently, our defense strategy focuses on identifying specific locations that significantly influence reconstruction. By applying a replacement algorithm, we aim to reduce the degree of data representation independence, thereby mitigating the effectiveness of such attacks.

In a reconstruction attack, adversaries obtain gradient information $Grad$ through backpropagation and then generate a random noise sample $X^*$ to approximate the original input $X$ using optimization methods $Invert(Grad)$. After applying our defense mechanism, the perturbed gradient is rep-

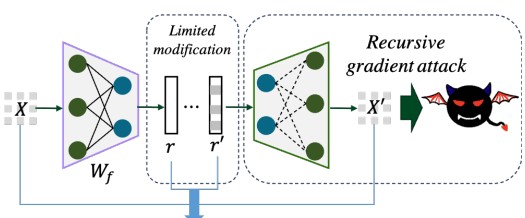

$maxmize \|X - X'\|$ , while $|r' - r|$ is limited

Figure 3: Illustration of the defense by data representation from intermediate output.

resented as $Grad^* = Defend(Grad)$, and the result obtained by the attacker from this perturbed gradient is $X' = Invert(Grad^*)$. To enhance privacy protection while minimizing costs and preserving model accuracy, we focus on selecting the data representations $r$ related to the gradient values most crucial for safeguarding privacy. Consequently, our optimization objective, illustrated in Fig. 3, is designed to maximize the distance between the original and reconstructed inputs, expressed as:

$$\textbf{Defense Goal: } max\|X - X'\|_p \tag{4}$$

Without loss of generality, we take the fully connected(FC) layer as an example to explore the rules related to the structure of fully connected layer networks and gradient computation. Let $X$ represent the input to the linear layer, and denote the output of $i_t h$ layer as $r^i$, $r^0 = X$. The output of the $(i + 1)_t h$ layer can be expressed as $r^{i+1} = W^i \cdot r^i$, where $W^i$ is the parameter matrix associated with $i_t h$ layer.

To achieve the defense objectives, we approximate the overall neural network as the mapping: $f : X \rightarrow r$, and utilize the inverse function to construct a reverse mapping from $r$ to $X$, thereby guiding modifications to $X$ through the calculation of $r$.

In this process, we choose those data representations $r$ that are more capable of influencing changes in $X$, using them to guide the replacement of model updates $\Delta g$ which adjacent to $r$ as the core of our replacement algorithm. As a result, even if attackers gain access to model update information, it becomes difficult to associate it with privacy training data through optimization or computational manner, thereby achieving the objective of privacy preserving. Subsequently, we make the following assumptions and present an inverse function theorem on guiding our algorithm to defense based on $r$.

**Assumption 1** *For $f : x \rightarrow y$, while $x \in R$, $f(x)$ is continuous: $\forall x_0 \in R, \lim_{x \rightarrow x_0} f(x) = f(x_0)$.*

**Assumption 2** *For $f : x \rightarrow y$, while $x \in R$, $f(x)$ is derivable: $\forall x \in R, \nabla f(x) = f'(x) = \lim_{\Delta x \rightarrow 0} \frac{f(x+\Delta x)-f(x)}{\Delta x}$.*

**Assumption 3** *For $f : x \rightarrow y$, there exists the inverse function, i.e. $f^{-1} : y \rightarrow x$, and the variation of $y$ is always bounded: $\forall y, y', \|y - y'\|_p \leq \epsilon$.*

**Theorem 1** *For feature extractor as the function $\exists f : X \rightarrow r$, based on Assumption 3, its inverse function is $f^{-1} = F : r \rightarrow X$. According to the conversion relationship, we have $\nabla f = (\nabla F)^{-1}$. $X$ represents the raw input and $r$ is the output vector of the middle layer. Our defense goal in Eq. 4 can be optimized as:*

$$X - X' = F(r_0) - F(r'). \tag{5}$$

In order to obtain an approximate solution, according to the Assumption 1 and 2, we perform the Taylor expansion at $r_0$ yields respectively,

$$f^{-1}(r_0) = F(r_0) = F(r_0) + F'(r_0) \cdot (r_0 - r_0) + \ldots \tag{6}$$

$$f^{-1}(r') = F(r') = F(r_0) + F'(r_0) \cdot (r' - r_0) + \ldots \tag{7}$$

Then we compute (6)-(7), and according to the trigonometric inequality transformation, we convert the result into the following formula:

$$\|(6) - (7)\| \approx \|F'(r_0) \cdot (r' - r_0)\| \geq \|\frac{r'}{f'(r_0)}\| - \|\frac{r_0}{f'(r_0)}\| \tag{8}$$

where $r_0$ represents the initial state after the forward calculation, $f'(r_0)$ represents the partial derivative calculated at the point, and $r'$ is the modified result. To maximize Eq. 8 and based on Assumption 3, the problem transforms into identifying the final position of $r'$ in a fine-grained manner, such that

$$\arg\max_{r'}\|\frac{r'}{f'(r_0)}\| \tag{9}$$

### 4.2.2 ADAPTIVE REPLACEMENT RATIO

Given the heterogeneous nature of private clients, including variations in model parameter sizes, it is necessary to set different replacement ratios for each client adaptively. To address this, we design an adaptive factor that determines the replacement ratio for each client based on local and historical information.

The client $C_k$ calculates the model update $\Delta g_k^t$ for the current iteration and derives the model parameters based on historical model information from the previous iteration $w_k^t$. Based on the ratio $\Delta g_k^t / w_k^t$, we can quantify the increment of the current model update relative to historical information as $A = \|\frac{\Delta g_k^t}{w_k^t}\|_1 / N$, which reflects the significant information captured during training and is positively correlated with privacy sensitivity. For such model updates, we will assign a higher replacement ratio. The client $C_k$ then determines its adaptive replacement ratio using the modified sigmoid function $sigmoid(x) = \frac{1}{1+e^{-x}}$, rescaled as $sigmoid \times 0.4 - 0.3$ to map the output from $\mathbb{R}$ to a constrained range of $(0, 0.1)$ as follows:

$$\mu_k^t = \frac{1 - 3e^{-A}}{10(1 + e^{-A})} + (1 - \beta) \tag{10}$$

where $\beta \in (0.1, 0.9)$ is an adjustable hyperparameter designed to adapt the proposed replacement algorithm to various datasets and applications, and $N$ denotes the size of the model parameters. The optimal setting of $\beta$ is verified through experiments. Fig. 2 shows the specific implementation of applying the above algorithms to the FL framework.

## 5 EXPERIMENTS AND ANALYSIS

### 5.1 EXPERIMENTAL SETUP

In this section, we conduct simulation experiments to verify the superiority of our proposed framework in terms of training accuracy and its effectiveness against privacy attacks.

**Datasets.** We select two widely used datasets, MNIST and CIFAR-10, to evaluate the effectiveness of our defense approach in real-world scenarios. The **MNIST** dataset is a benchmark for machine learning tasks, containing 70,000 gray scale images of size $28 \times 28$. The **CIFAR-10** dataset is a widely recognized benchmark for image recognition tasks. It consists of $32 \times 32$ RGB images across 10 classes, including animals and vehicles. We adopt the non-IID partitioning of the dataset, consistent with our hypothesis that data held by different clients exhibit varying label distributions. The dataset is divided into 10 disjoint parts, each allocated to one of the 10 clients in the FL scenario. One partition is designated as the auxiliary dataset and used to train the guard client, ensuring that its data distribution remains uncorrelated with the other private clients.

**Hyperparameters configurations.** We employ the LeNet-5 model architecture following Zhu & Han (2020). For the MNIST dataset, the learning rate is set to 0.01, with 10 local epochs, 50 global epochs, and a batch size of 256. For CIFAR-10, the learning rate is 0.001, with 20 local epochs, 50 global epochs, and a batch size of 64. The model is optimized using the *SGD* optimizer and Cross-Entropy loss function.

**Privacy Attack.** (1)*Reconstruction Attack:* For comparability, we follow the setup in the GS Attack(Geiping et al. (2020)). During the privacy-preserving test phase, we employ a local batch size of 1, which is the simplest and most effective configuration to reconstruct training samples from the shared model updates. (2)*Membership inference Attack:* We employ the Boundary Attack(Li & Zhang (2021)), a decision-based inference method that does not require shadow models or datasets. Sensitivity of the local models is assessed using 500 member samples from the training data and 500 non-member samples from external sources.

**Evaluation Metrics.** To validate the effectiveness of our proposed scheme, we aggregate local model updates following our defense scheme to assess its impact on model accuracy. To evaluate the scheme's robustness against privacy attacks, we use Mean Square Error (MSE) and Peak Signal-to-Noise Ratio (PSNR) to quantify the difference between the reconstructed image and the original image—both metrics commonly used for reconstruction tasks. Additionally, we measure the AUC of the attack accuracy for the membership classifier, illustrating the framework's defense capability.

## 5.2 MODEL PERFORMANCE

### 5.2.1 COMPARISON WITH THE BASELINE

We first evaluate the effectiveness of our framework by training global models on the MNIST and CIFAR-10 datasets, as shown in Fig. 4a. The results indicate that although the convergence speed of the proposed method is slightly reduced, it still achieves a high level of accuracy, comparable to the baseline. As the adaptive factor decreases and the replacement ratio increases, the convergence accuracy initially stabilizes at an optimal value but declines once the threshold is exceeded. Based on empirical analysis, $\beta = 0.7$ is identified as a suitable hyperparameter for the MNIST scenario and is adopted for comparison with other methods. And our method is also effective for the cifar10 dataset.

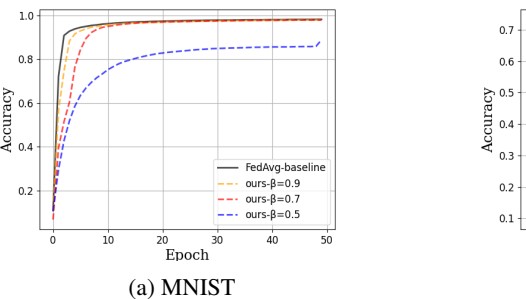 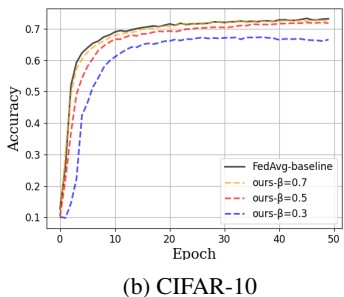

(a) MNIST                    (b) CIFAR-10

Figure 4: Accuracy comparison on MNIST and CIFAR-10 datasets, respectively.

### 5.2.2 COMPARISON WITH THE STATE-OF-THE-ART

To demonstrate the superiority of our approach, we compare it with two state-of-the-art defense methods. The method **POGZ-FL** proposed in Zhu et al. (2022) calculate the importance value of each layer to reallocate the privacy budget. Miao et al. (2022) propose a method called **CA-FL** which combines compressive sensing with differential privacy.

We compare the proposed method with the aforementioned state-of-the-art defense methods on the MNIST dataset, and the results are shown in Fig. 5a. The parameter $\alpha$ for POGZ-FL is set to 1.0, while $\epsilon$ for CA-FL is set to 5, with a compression ratio of 0.1, as recommended in their respective papers for optimal accuracy. The adaptive value $\beta$ for our method is set to 0.7.

From the results, we observe that although POGZ-FL exhibits a faster initial convergence compared to our method, it lacks stability and fails to achieve final convergence. CA-FL, on the other hand, significantly slows its convergence due to simultaneous gradient compression and adaptive differential privacy operations. In contrast, while our proposed method shows a slightly slower convergence rate compared to the baseline, it ultimately achieves accuracy comparable to the baseline, demonstrating its superior practicality.

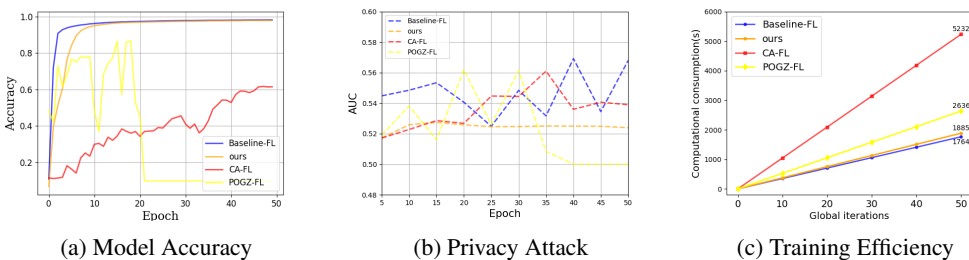

|  | (a) Model Accuracy | (b) Privacy Attack | (c) Training Efficiency |

Figure 5: Comparison of different indicators under four defense frameworks.

|  | Ground truth | Initial noise | No defense | Ours | CA-FL | POGZ-FL |
|---|---|---|---|---|---|---|
| [Scenario 1]
MNIST
Label '2' | \ | \ | MSE=0.0262
PSNR=15.82 | MSE=0.4277
PSNR=3.69 | MSE=0.1835
PSNR=7.3628 | MSE=0.3339
PSNR=4.7632 |
| [Scenario 2]
MNIST
Label '4' | \ | \ | MSE=0.0035
PSNR=24.59 | MSE=0. 9835
PSNR=1.02 | MSE=0.5881
PSNR=2.31 | MSE=0.4481
PSNR=3.49 |
| [Scenario 3]
CIFAR-10
Label 'airplane' | \ | \ | MSE=0.0110
PSNR=19.5982 | MSE=4.4469
PSNR=-6.48 | MSE=4.0600
PSNR=-6.09 | MSE=3.3593
PSNR=-5.26 |
| [Scenario 4]
CIFAR-10
Label 'ship' | \ | \ | MSE=0.0323
PSNR=14.91 | MSE=0.9296
PSNR=0.32 | MSE=0.3598
PSNR=4.44 | MSE=0. 2617
PSNR=5.82 |

Figure 6: The effectiveness of various defense mechanisms against GS attack in different scenarios.

## 5.3 Privacy Preserving

### 5.3.1 Defense for Reconstruction Attack

To evaluate the privacy-preserving effectiveness of the proposed method, we conduct GS attacks, a more generalized and potent gradient inversion attack compared to the DLG(Zhu & Han (2020)). GS attacks utilize both the gradient and prior knowledge of the dataset, such as mean and variance, for more accurate reconstruction. We compare the performance of our method against CA-FL and POGZ-FL under GS attacks, as shown in Fig. 6.

The GS attack is applied to an untrained LeNet5 network with weights initialized from a uniform distribution, using noisy samples as dummy data. To ensure experimental reliability, we conduct five trials and analyze the best result.

Without any defense, the GS attack achieves nearly complete reconstruction, with an MSE below 0.05 and images closely resembling the ground truth.Our defense mitigates this by replacing the target gradient with a guard gradient from randomly selected samples, yielding an MSE of 0.9835, making the reconstructed image unrecognizable. We also test the CA-FL and POGZ-FL defense methods, using their respective hyperparameters as outlined in 5.2.2. On the MNIST dataset, although these methods can resist GS attacks, they still cause partial recognition of features, with lower MSE values compared to our method. On the CIFAR-10 dataset, where the complexity of the scenario increases, the attack's reconstruction accuracy generally decreases, but our method continues to provide a higher MSE, indicating superior privacy protection.

### 5.3.2 Defense for Membership Inference Attack

In this experiment, we implement the Boundary Attack which is a decision-based membership inference attack to assess the privacy-preserving effectiveness of various defense methods. Specifically, we iteratively apply QEBA perturbations Li et al. (2020) to each correctly inferred sample until the model changed its decision, using the L2 distance of the perturbation as the criterion for determining membership. If the perturbation exceeds a predefined threshold, the sample is classified as a member. The Area Under the Curve (AUC) value is then calculated to evaluate the effectiveness of the attack.

For the attack, we randomly select a local model as the victim every five global iterations. The attack's effectiveness is evaluated across four different frameworks, with the results presented in Fig.5b.

The results indicate that as global iteration rounds increase, the accuracy of the membership inference attack does not change significantly but increases slightly. This suggests that the federated learning approach helps mitigate overfitting across different clients, though there remains room for improvement. Specifically, the baseline method achieved an average attack AUC of 0.5453, while POGZ-FL and CA-FL attained AUCs of 0.5235 and 0.5357, respectively. Our method achieved an average AUC of 0.5245, representing a 4.0% reduction compared to the baseline method.

### 5.4 Efficiency

To assess the efficiency of various defense methods, we measure the average training time of the model under four conditions.This comparison highlights the communication and computation delays introduced by each defense method in the federated learning scenario. By maintaining consistent experimental parameters and hardware, we focus solely the impact of incorporating different defense mechanisms on training time. The experimental results are shown in Fig. 5c.

With 50 global iterations, the method without defense takes approximately 1764 seconds, while our defense method requires only 1885 seconds, reflecting a minimal 6.8% increase. In comparison, the POGZ-FL method takes 2636 seconds, a 49.4% increase due to the computational overhead of calculating differential privacy coefficients and adding noise. Although CA-FL aims to reduce noise overhead through model compression, the added cost of compression and decompression raises training time to 5232 seconds. Thus, our method demonstrates superior efficiency.

## 6 Conclusion

In this work, we present the observation that the data representations of a model's intermediate outputs are independent of one another. Building on this insight, we propose a replacement algorithm that leverages data representations with varying distributions for entanglement, applying it to federated learning scenarios. This approach ensures model accuracy while enhancing resistance to privacy attacks. Specifically, we introduce the guard client whose updated information is used to replace and fuse as the base with private client data before uploading to the server, thereby optimizing collaboration among clients within the federated learning framework. Furthermore, we provide a theoretical analysis and discussion supporting the use of data representations for replacement. Finally, we conduct extensive experiments to validate the effectiveness of our method in both global model accuracy and resisting attacks. The results demonstrate that our approach offers significant improvements in terms of accuracy, privacy preserving, and efficiency compared to other widely-used privacy-preserving techniques.

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

# A    APPENDIX

## A.1    DISCUSSION ON ADVERSARY BASED ON DATA REPRESENTATIONS

In the context of mini-batch training and the optimization of the stochastic gradient descent algorithm, using a single training sample can lead to data representation independence in the model's intermediate computation results, which we believe is the root cause of privacy attacks.The feature differences of data representations have been exploited in various privacy attacks.In reconstruction attacks, Zhu & Blaschko (2021) can calculate the original data input by leveraging data representations calculations when the model structure is known. In membership inference attacks, data representations can also serve as effective features for constructing the attack classifiers.

Based on the concept of data representation independence and supporting research, it has been demonstrated that an attacker can utilize recursive algorithms to infer the intermediate data representations of an entire neural network when the gradient is known, thereby enabling a reconstruction attack. Taking the example of the neural network with fully connected structures describes as:

$$z = W_d f_{d-1}(x) \tag{11a}$$
$$f_{d-1}(x) = \sigma_{d-1}(W_{d-1} f_{d-2}(x)), \tag{11b}$$

while $d$ denotes the d-th layer, $\sigma$ denotes the activation function and $f_{d-1}$ represents the model structure before the $d-1$ layer. According to the Eq.11, we can calculate the gradients according to

the following chain rule:

$$\frac{\partial l}{\partial W_d} = \frac{\partial l}{\partial z} f_{d-1}^T \tag{12a}$$

$$\frac{\partial l}{\partial W_{d-1}} = ((W_d^T(\frac{\partial l}{\partial z})) \odot \sigma_{d-1}') f_{d-2}^T \tag{12b}$$

$$\frac{\partial l}{\partial W_{d-2}} = ((W_{d-1}^T((W_d^T(\frac{\partial l}{\partial z})) \odot \sigma_{d-1}')) \odot \sigma_{d-2}') f_{d-3}^T \tag{12c}$$

We observe that the gradient of each layer has a repetitive format and is dependent on the output of the previous layer $f(x)$. It is possible to calculate the neuron outputs of each layer in reverse, starting from the output $z$ of the final layer. Specifically, when the gradient is known, the neuron outputs of each layer can be computed in reverse, starting from the output $z$ of the last layer until the original input $x$. In this process, we found that due to the rules of chain computing, the output results of the intermediate layer are directly related to the gradient values. For privacy attackers, these key neuron outputs are critical, as they significantly impact the accuracy of the reconstructed data and, consequently, the success of the final reconstruction. Therefore, modifying the gradient at these critical points can effectively protect private data from optimization-based reconstruction attacks.

Thus, by understanding the theory of data representation independence, we can identify specific locations in the model that are vulnerable to privacy attacks. Our approach involves using a gradient replacement algorithm to deliberately entangle data representations, thereby resisting privacy attacks.

