# OpenReview forum: "Model Entanglement for solving Privacy Preserving in Federated Learning"
_ICLR.cc/2025/Conference — Submitted to ICLR 2025_

### Official Review · Reviewer_kEWo · 2024-10-31

**Soundness:** 1
**Presentation:** 1
**Contribution:** 2
**Rating:** 3
**Confidence:** 5

**Summary:**

This paper proposes to use model entanglement with a guard model mechanism to mitigate the privacy issue in federated aggregation.

**Strengths:**

This work studies the fundamental privacy problem in federated learning during client update aggregation, where finding a balance between utility and privacy is still a challenging issue at the moment.

**Weaknesses:**

1. The idea of model entanglement only applies to the model structure of convolutional layers and fully connected layers, which is extremly limited considering other privacy-preserving FL solutions based on primitives like HE MPC, and DP can easily handle abitrity types of NNs. Additionally, the paper fails to mention HE-based and TEE-based solutions, which are two of the most promising PPFL solutions currently.
2. There is no rigorous security proof in the paper. One of the core challenges of PPFL is to provide a formal privacy guarantee. It seems like the proposed solution will not be able to provide such proof with obfuscation as its core. On the contrary, other privacy primitives can easily achieve that, e.g., HE with computationally indistinguishability and DP with statistical guarantee. Also, the threat model is not well defined (rather a laundry list of two popular categories of practical privacy attacks), which in turn cannot clearly capture the privacy guarantee here. For example, since the guard model described in the paper is trained on public data, an adversary with sufficient computing power and enough auxiliary information can also train a similar model to help with its attacks.
3. The paper only exmained a simple LeNet in the experiments and the defense results only show a few hand-picked samples without a more thorough analysis of a larger-scale. The experiements seem to be merely a simulation, which is not sufficient in the setup of FL where pratical systems will experience issues like client dropout which would impact the defense.
4. The writing of this paper is subpar. For example, it is hard to follow the logical transition from the observation to the proposed model entanglement solution in Section 4.1. Additionally, there are several grammar issues and typos including "privacy preserving by obfuscating" at Line 19, "Membership Inference Attack(MIA)" at Line 73, "a interfernece" at Line 253, etc. The figures in the paper are in low resolution when zoomed in for details. One suggestion is to use pdf or svg.

**Questions:**

1. The idea behind model entanglement is based on label-level intermediate representation, how do you guarantee the privacy for more fine-grained inversion attacks with stronger computing power that seems to be able to bypass your defense here?
2. Is there any more theoritical proof on how the guard model updates impact model training, e.g. model convergence?
3. Could you explain the detailed setup of Figure 5b? Why do you think randomly selecting a local model as the victim every five global iterations is enough of a privacy indicator? (In general, security and privacy are about the worst-case scenario).
4. How does the adaptive replacement ratio affect model accuracy and privacy preservation, and how do you determine the optimal replacement ratio for different datasets and client distributions?

---

### Official Review · Reviewer_nATq · 2024-11-03

**Soundness:** 2
**Presentation:** 2
**Contribution:** 2
**Rating:** 3
**Confidence:** 4

**Summary:**

The authors point out a key observation that the middle layer outputs of a neural network can exhibit independence in value distribution across different types of data. Then the authors propose a ME method to improve the balance between privacy preservation and model accuracy.

**Strengths:**

1. The paper points out a key finding about middle layer outputs.
2. The paper propose a defense method for membership inference attack and model inversion attack, and use some experiments to verify it.

**Weaknesses:**

1. I expect more theoretical analysis about the key findings, as the finding about middle layer data representation ability seems to be trivial.

2. The experimental results are not robust. 1) The paper claims to enhance the trade-off between utility and privacy, yet it does not present the trade-off curve; 2) The paper should compare additional privacy-preserving methods in FL; 3) The privacy attack mentioned, such as that by Geiping et al., is not state-of-the-art; 4) A larger dataset and neural networks are necessary.

3. The paper writing needs to be improved. For example, in theorem 1, I did not find a clear theorem claim.

**Questions:**

Question
1. Could you please given more theoretical analysis about the finding? Especially in different neural network architectures.
2. Could you please provide more experimental evidence to validate the defense and model performance.

---

### Official Review · Reviewer_G2vP · 2024-11-03

**Soundness:** 3
**Presentation:** 2
**Contribution:** 2
**Rating:** 5
**Confidence:** 3

**Summary:**

The paper studies the privacy leakage of the client's training examples via communicated gradients/model updates in federated learning. The authors notice that the intermediate representations of each layer have strong correlation to the input data and hence one can identify weights that contain more information of the training data by looking at the representations. Using this method, the paper proposes to replace the identified weights with ones that are trained on a public dataset which is considered to be privacy-preserving. Empirical experiments on MNIST and CIFAR-10 show that this method decreases the effectiveness of membership inference attacks and reconstruction attacks, without hurting the performance of the overall FL algorithm. Compared to other defenses, experimental evaluation shows that the defense is often more effective, more accurate and more computationally efficient.

**Strengths:**

* Privacy in FL is an interesting and important topic
* Intuitive defense method to replace the more sensitive parameters with ones trained with public data
* According to the experimental results, the proposed defense is more effective and efficient than existing ones without significantly hurting the FL performance

**Weaknesses:**

* **Lacks formal privacy guarantees.** In the introduction, the authors claim that they are motivated by the lack of privacy guarantees in FL. However, the proposed method also gives no privacy guarantee.
* **The assumption of public data.** The proposed defense is only feasible if one have access to a client that trains on public data to provide a reference model. While I agree that it is usually easy to find public data that is similar to other clients' data, in the experiment, the public data is just a partition of the dataset like the data of all other clients. More often, the public data one can obtain is more general than the clients' private data. For example, when using CIFAR-10 as clients' data, we can use ImageNet or CIFAR-100 as the public dataset. I suggest the authors evaluate the defense on this setting.
* **Algorithm description.** Algorithm description lacks clarity and could benefit from a pseudocode.
* **Insufficient experiments.** The experiments are limited to two tiny image datasets. In Fig. 6, the reconstruction attack results are based on only 4 examples. It would be nice to run the attack on more examples and report the overall mean and std of the attack MSE with the proposed defense and other defense methods. Also, it would be helpful if the results with DP-FedAvg are provided as a private baseline that achieves guaranteed privacy.
* **Insufficient description of baseline defense methods.** More details on how they work will be appreciated.
* **Presentation.** There are some typos and grammatical errors. Please proof read the paper and make edits accordingly.

**Questions:**

* How would the defense perform with more realistic public datasets (e.g., using ImageNet/CIFAR-100 as public data for CIFAR-10 private data)?
* Please clarify the layer selection process: Does the algorithm consider all layers (convolutional, fully connected) when choosing weights to replace?
* How does weight replacement affect the local model's performance on its own data distribution? This is particularly interesting as complete replacement with the reference model would achieve perfect privacy but would destroy local utility (not so much for global utility though).
* Do the other defense methods compared in the paper also make use of public data? If not, it may not be a fair comparison.

---

### Official Review · Reviewer_6k74 · 2024-11-04

**Soundness:** 2
**Presentation:** 1
**Contribution:** 1
**Rating:** 1
**Confidence:** 2

**Summary:**

The paper presents a defense mechanism against Membership Inference Attacks and Model Inversion Attacks in the Federated Learning setup. The proposed defense strategy is based on obfuscating intrinsic representations. The authors provide theoretical foundations for their method and conduct experimental comparisons against several baseline approaches.

**Strengths:**

The chosen research direction of privacy preservation in Federated Learning represents a significant and timely area of investigation.

**Weaknesses:**

1. The paper exhibits significant issues with writing quality, which impedes comprehension. The manuscript contains numerous typographical errors, spacing problems, and grammatical inconsistencies that seriously compromise its readability. For example: line 38 "Zhu et al", line 38 "work", lines 290 and 291 "i-th" and so on.

2. (Crucial) The proposed method lacks clear and precise description. While Sections 4.2.1 and 4.2.2 attempt to explain the algorithm's operational principles, they fail to explicitly specify the method of selecting data representation for further obfuscation. Furthermore, Section 4.1 provides insufficient clarity regarding the specific implementation of the auxiliary dataset and guard model, and their relationship with representation obfuscation. The interaction between clients illustrated in Figure 2 lacks corresponding explanation. The authors should consider to add detailed pseudocode of the algorithm to the paper.

3. (Crucial) The mathematical formulation demonstrates concerning imprecision and inconsistencies. For example:
- Assumption 3 fails to establish a clear relationship between the variation of $y$ and the invertibility of function $f$. The condition $\|y - y'\|_p\leq \epsilon$ lacks proper definition. Does it mean that all considered $y$ values lie within an $\epsilon$-ball? Or does it suggests the existence of an inverse mapping $f^{-1}$ for some $\epsilon$-ball centered at $y$?
- The space in which $y$ (or $r$, for some reason, the authors changed $y$ to $r$ for the Theorem 1) exists is not properly defined. Since in Equations 6 and 7 the authors use $\cdot$ for the multiplication $F'(r_0)\cdot(r'-r_0)$, one could suggest, that $r\in\mathbb{R}$. However, Equation 8 uses the norm of $r'$, which is more suitable for multivariate case. Moreover, the one-dimensional $x$ treatment in Assumption 1 (assuming $R$ denotes $\mathbb{R}$; if not, R remains undefined) contradicts the multivariate nature of $X$ and $r$.
- The relationship between Equations 4 and 5 is ambiguous. While Equation 4 presents an optimization problem (with undefined optimization variables), Equation 5 states an identity without clear connection to the preceding equation.

4. The paper exhibits significant notation inconsistencies and undefined variables. For example:
- The variable $T$ in Equation 3a lacks definition
- The role of $\mu$ in Equation 10 remains unexplained
- The symbol $N$ is inconsistently used, initially defined as the number of clients (line 130) and later redefined as the number of parameters (line 355)

Summing up 3. and 4. the authors could consider to include a notation table or glossary to the paper to clearly define all variables and symbols.

5. The paper demonstrates imprecise use of terminology and abbreviations. E.g.:
- Section 4.2.2 introduces "replacement ratios" without prior definition or context
- The abbreviation DLG appears on line 113 without previous introduction or explanation

6. (Crucial) The experimental setup and results lack necessary detail and clarity:
- The specific methodology for dataset distribution among clients is not adequately described
- Section 5.2 fails to specify which particular attack was being evaluated
- The attack implementations lack comprehensive description and technical details

Paper would benefit from providinf a detailed experimental protocol (and, ideally, the code), including specifics on dataset distribution among clients.

7. Figure 5(c) and Section 5.4 present results on the Efficiency of the considered algorithms. However, the relative efficiency of different algorithms can significantly vary depending on the specific setup and hardware used. The authors provide no such description, limiting themselves to just "consistent experimental parameters and hardware"; they also do not provide a Reproducibility Statement and experimental code, which could have revealed more details about their experimental setup.

**Questions:**

See Weaknesses (except 1).

---

### Meta-Review · Area_Chair_kkwQ · 2024-12-17

**Metareview:**

The reviewers raised several major concerns about the paper, including the poor presentation and lack of rigor both in terms of the mathematical formulation and the experimental evaluation. The author(s) did not provide any response during the rebuttal phase. This submission is well below the acceptance bar at ICLR. We strongly encourage the author(s) to address the concerns raised by the reviewers before resubmitting the work to a conference.

**Additional Comments On Reviewer Discussion:**

The reviewers raised several concerns about this work:
1) The poor writing quality
2) The lack of a clear and precise description of the proposed method
3) The imprecise and inconsistent mathematical formulation
4) The unclear experimental setup
5) The lack of formal privacy guarantees

No response was provided by the author(s).

---

### Decision · Program_Chairs · 2025-01-22

Reject